# First Data on PAE Levels in Surface Water in Lakes of the Eastern Coast of Baikal

**DOI:** 10.3390/ijerph20021173

**Published:** 2023-01-09

**Authors:** Selmeg V. Bazarsadueva, Vasilii V. Taraskin, Olga D. Budaeva, Elena P. Nikitina, Svetlana V. Zhigzhitzhapova, Valentina G. Shiretorova, Tcogto Zh. Bazarzhapov, Larisa D. Radnaeva

**Affiliations:** 1Baikal Institute of Nature Management, Siberian Branch of the Russian Academy of Sciences, Sakhyanovoi St., 6, 670047 Ulan-Ude, Russia; 2Department of Pharmacy of the Medical Institute, Banzarov Buryat State University, 670000 Ulan-Ude, Russia

**Keywords:** lakes of the eastern coast of Baikal, PAEs, spatiotemporal variation, source of origin, risk assessment

## Abstract

The increasing consumption of phthalates (PAEs), along with their high toxicity and high mobility, poses a threat to the environment. This study presents initial data on the contents of six priority PAEs in the water of lakes located on the eastern shore of Lake Baikal-Arangatui, Bormashevoe, Dukhovoe, Kotokel, and Shchuchye. The mean total concentrations of the six PAEs in lakes Arangatui and Bormashevoe (low anthropogenic load) were comparable to those in Kotokel (medium anthropogenic load, 17.34 µg/L) but were significantly higher (*p* < 0.05) than in Dukhovoe and Shchuchye (high anthropogenic load, 10.49 and 2.30 µg/L, respectively). DBP and DEHP were the main PAEs in all samples. The DEHP content in lakes Arangatui and Bormashevoe was quite high, and at some sampling sites it exceeded the MACs established by Russian, U.S. EPA, and WHO regulations. The assessment showed that there is no potential risk to humans associated with the presence of PAEs in drinking water. However, the levels of DEHP, DBP, and DnOP in the water pose a potential threat to sensitive aquatic organisms, as shown by the calculated risk quotients (RQs). It is assumed that the origin of the phthalates in the studied lakes is both anthropogenic and biogenic.

## 1. Introduction

The pollution of ecosystems with plastic waste and phthalates, derivatives of phthalic acid, has been a global environmental problem. PAEs are used in almost all industries, in building materials (PVC, flooring, etc.), printing inks, varnishes, latex inks, cosmetics, clothing, food packaging, pharmaceutical preparations, medical products, insecticides, etc. Finished plastic products, such as plastic films, medical devices, and tubes, may contain up to 20–60% PAEs by weight [1,2,3,4,5]. The World Health Organization (WHO) classifies PAEs as endocrine disruptors [6]. The toxicity of PAEs is due to their ability to bind to the human hemoglobin molecule and affect red blood cell function [7], leading to DNA damage [8] and pregnancy termination [9]. Some PAE congeners such as di-(2-ethylhexyl) phthalate (DEHP) and butyl benzyl phthalate (BBP) are considered to be potentially carcinogenic to humans [10]. It is worth noting that the combined effects of PAEs and other pollutants on living organisms have both cumulative and antagonistic effects [11,12].

Today, largely thanks to the efforts of the scientific community, the state pays considerable attention to the ecological condition of Lake Baikal, a unique natural system, the world’s reserve of fresh water, and a UNESCO World Heritage Site. However, PAE monitoring in Lake Baikal and its catchment area is not included in the monitoring programs of state agencies, such as Rosgidromet, the Ministry of Natural Resources and Environment of Russia, and others. Back in 1996–2002, the DEHP content in the water of Baikal was found to be 0.09–1.40 µg/L, and the DEHP content in its tributaries was found to be 0.10–0.50 µg/L [13,14]. In recent years, the problem of PAE pollution in Lake Baikal has been exacerbated by the occurrence of microplastic in the lake. Some authors have noted a high degree of plastic pollution in the lake [15]. In a recent study, the DBP and DEHP contents in Baikal water were determined to be 0.35–0.89 and 0.06–0.32 µg/L, respectively. A simultaneous change in the DBP concentration in the surface water layer in the pelagic zone of the southern Baikal depression, as well as in its coastal zone, serves as an indicator of pollution [16]. At the same time, the origin of PAE pollutants is apparently not always anthropogenic: works have proven a biogenic origin as well [17,18,19,20,21].

Water of lakes, along with rivers and groundwater, plays an essential role in the water economy and is the most important surface water resource. Lake ecosystems are characterized by slower water exchange and can therefore accumulate pollutants, in some cases posing a danger to living organisms and humans. Large lakes may be of greater interest to researchers than small ones due to their potential for water supply, fisheries, navigation, etc. In contrast, small lakes are generally poorly studied, although small lakes are known to respond more quickly to anthropogenic loads and environmental changes. Consequently, changes in the ecosystems of small lakes are more pronounced, and their chemical compositions are more susceptible to local influences. No studies of PAEs in the lake ecosystems of the Baikal natural area have been conducted. In view of the above, a study of the PAE pollution levels in aquatic ecosystems is necessary and timely, as is the determination of their sources and their methods of transformation in components of the aquatic ecosystem as well as an assessment of the associated environmental risks. These data can be used to prepare scientific guidelines and environmental regulations to prevent PAE pollution.

For the first time, this paper reports initial data on the contents of six priority PAEs in the ecosystems of large lakes (Arangatui, Bormashevoe, Dukhovoe, Kotokel, and Shchuchye, located on the eastern shore of Lake Baikal) that have varying degrees of anthropogenic load. The obtained results confirm the complex formation of the phthalate pool in the water of the lakes and may represent important information about the distribution of PAEs in the Baikal natural area.

## 2. Materials and Methods

### 2.1. Materials and Reagents

Standard PAE compounds, such as dimethyl phthalate (DMP), diethyl phthalate (DEP), dibutyl phthalate (DBP), benzyl butyl phthalate (BBP), di-n-octyl phthalate (DnOP), and di-(2-ethylhexyl) phthalate (DEHP), as well as the deuterated surrogate standards DMP-d4, DEHP-d4, and an EPA Phthalate Esters Mix were procured from Sigma-Aldrich (Burlington, MA, USA). All the solvents used (methanol, acetone, ethyl acetate, methylene chloride, and n-hexane) were pesticide- or HPLC-grade. For sample filtration, we used glass fiber filters with a pore size of 0.45 μm without binders produced by Jiuding High-Tech Filtration, Beijing, China. The EPA Phthalate Esters Mix (Accustandard Inc., New Haven, CT, USA) included 2000 μg/mL of each component (DMP, DEP, DBP, BBP, DEHP, and DnOP). Standard solutions were prepared by diluting the solution in n-hexane. All solutions were stored in amber glass bottles at 2 °C in a refrigerator. The extraction of target analytes was performed on C18 SPE ENVI-18 cartridges (6 mL, 500 mg, Supelco, Waltham, MA, USA) using an SPE Vacuum Manifold VM12 (Phenomenex, Torrance, CA, USA) (Appendix A).

The eluates were dehydrated with anhydrous sodium sulfate (Lenreaktiv, Saint-Petersburg, Russia).

Laboratory equipment made of glass, stainless steel, and polytetrafluoroethylene was used for sampling and experiments. All glassware was washed with hot tap water without detergents, then successively washed with diluted sulfuric acid (H_2_O:H_2_SO_4_ = 2:1 by volume), bidistilled water, acetone, methylene chloride, and hexane. The glassware was then dried at +400 °C for 4 h (except for volumetric ware) and washed again with methylene chloride and hexane.

### 2.2. Study Area and Sampling

Three groups of lakes with low, medium, and high anthropogenic loads were selected as objects of research: I—Arangatui and Bormashevoe, II—Kotokel and Dukhovoe, and III—Shchuchye.

Lakes Arangatui, Bormashevoe, Dukhovoe, and Kotokel are located in tectonic depressions formed by rift processes within 3 km of the coast of Lake Baikal. Their average depths range from 2 to 6 m, and the area varies from 1.3 to 62.9 km^2^. The chemical composition of the water in these lakes is significantly influenced by the discharge of thermal fracture-vein waters. Two lakes—Shchuchye and Kotokel—have long been intensively used as recreational water bodies where dozens of camping sites and guest houses are located. The territory where the lakes are located is characterized by a sharply continental climate. The average annual temperatures are negative and range from −3 °C to −2 °C, with a short period of frost-free days (95–100 days) and relatively low precipitation, ranging from 200 to 350 mm (50% of which falls in July and August, with only 15% in the cold season). Due to severe cold winters (the average monthly air temperature remains negative for 6 months), lakes freeze to a considerable depth (ice thickness up to 1.2 m), and during the warm season there is a short ice-free period. Detailed information about the geographical and hydrological characteristics of the studied lakes and anthropogenic load data are provided in the Appendix A. The selected lakes are convenient models for the comparative analysis of phthalate pool formation under different conditions.

Surface water samples were taken in February 2022 (winter low water, dry season), when lakes are covered with ice and tributaries are frozen over. During the subglacial period, there was a lack of dissolved oxygen in the water and there was no UV radiation; the processes of bacterial activity were slowed down, which led to increased degradation time for PAEs. It was expected that under these conditions the PAE concentrations in the water should reach their maximum values.

At each lake, surface water samples were taken from 3–5 sites. A map with the studied lakes and sampling sites is shown in Figure 1.

Surface water samples were taken in three parallel 1 L amber glass bottles. Then, 1 mL of HNO_3_ was added to each sample for preservation, and the samples were placed in refrigerators and immediately transported to the laboratory. The PAE concentrations were measured within 24 h of sample receipt.

### 2.3. Sample Pretreatment

The samples were pretreated according to EPA methods 3535 and 8061A, with slight modifications. The water samples (500 mL) were filtered through a 0.45 µm fiberglass membrane in a glass filter unit. The PAEs were concentrated by solid phase extraction (SPE) on ENVI-18 cartridges (500 mg, 6 mL, Supelco, USA) using a VM12 vacuum manifold (Phenomenex, USA), and were sequentially conditioned with 5 mL of acetone, 5 mL of methylene chloride, 5 mL of ethyl acetate, 5 mL of acetone, and 5 mL of purified water. The cartridges were then dried for 20 min, and the PAEs were sequentially eluted with 6 mL of methylene chloride and 6 mL of n-hexane at a rate of 1 mL/min [21]. The solvents from the eluate were removed under vacuum using a rotary evaporator to a volume of 1 mL, then evaporated almost to dryness under a weak flow of nitrogen. The residue was dissolved in 1 mL of n-hexane for analysis using gas chromatography–mass spectrometry (GC/MS).

### 2.4. GC–MS Analysis

A GC/MS analysis of phthalates was performed on an Agilent 7890B GC in tandem with an Agilent 7000C MS in selective ion monitoring (SIM) mode, according to US EPA method 8270D, with slight modifications. An Agilent HP-5MS UI capillary column (30 m × 0.25 mm × 0.25 μm) was used with 6.0 grade helium (He 99.9999%) as a carrier gas at a flow rate of 1 mL/min. The oven temperature program was as follows: 60 °C for 1.0 min; to 220 °C at 20 °C/min and hold for 1.0 min; to 280 °C at 5 °C/min and hold for 2.0 min. Injection was in split mode (5:1), with an injection volume of 1 μL, an injection port temperature of 280 °C; and an interface temperature of 280 °C in electronic ionization (EI) mode with an electron energy of 70 eV (Appendix A).

An appropriate concentration of a standard solution was spiked into each sample to estimate the recovery and the performance of the methods. A procedural blank and a triplicate sample were determined in each batch of samples. PAE concentrations were adjusted for spiked, procedural, and solvent blanks.

### 2.5. Quality Assurance and Quality Control

To ensure the accuracy and usefulness of the laboratory data, the instruments were proofread using calibration standards, and spiked, procedural, and solvent blanks were carried out for every set of 16 samples or new test day. The correlation coefficients for the six calibration curves were greater than 0.98% (Appendix A).

The minimum detection limit (MDL) was estimated using the 3σ IUPAC criterion (the signal-to-noise ratio ≥ 3) and was in the range of 0.10–0.70 ng/L. The minimum quantification limit (MQL) was estimated using the 10σ IUPAC criterion (the signal-to-noise ratio ≥ 10) and was within 0.30–2.10 ng/L.

The final concentration values were calculated by subtracting the mean values of the procedural blanks, and values less than the MQL were considered to be zero. The recovery rates ranged from 78.20% to 118.15%. Additional information about the method, such as the reproducibility, accuracy, etc., are given in Appendix A.

### 2.6. Analysis of Water Quality Parameters

The temperature, turbidity, pH value, dissolved oxygen (DO), salinity, content of phosphates, ammonium, nitrites, nitrates, total phosphorous (TP), chemical oxygen demand (COD), and permanganate index (PI) in the water were measured in a field laboratory using additional equipment—a pH tester (Hanna portable instruments; HI 991300 and HI 98703) and a photoelectric colorimeter (PE-5400 UV, Ecroskhim, Saint-Petersburg, Russia)—on the day of sampling. The water pH was measured using the potentiometric method, the DO content was measured using the Winkler test, and the mass concentration of nitrites was measured by photometric detection using a Griess reagent. The nitrate concentration was determined with salicylic acid using the photometric method, the concentration of ammonium ions was determined by photometric detection using Nessler’s reagent, and the phosphate concentration was determined by photometric detection through ascorbic acid deoxidation. Nutrients were analyzed using a photocolorimeter. The COD was determined based on the oxidation of organic substances by excess potassium dichromate in a sulfuric acid solution in the presence of a catalyst (silver sulfate) and subsequent photometric detection. The PI was determined with the back titration method using a standard oxalate solution. The water quality parameters of the lakes on the eastern shore of Lake Baikal are presented in Appendix A.

### 2.7. Ecological Risk Assessment

#### 2.7.1. Human Health Risk

The exposure to PAEs for local populations as well as risk levels were calculated using Equation (1) [22]:(1)$$\mathrm{AE}=\frac{\mathrm{Cw}\times \mathrm{IR}\times \mathrm{EF}\times \mathrm{ED}}{\mathrm{BW}\times \mathrm{AT}},$$
where AE is the level of exposure of an adult to PAEs when drinking water, mg/kg of body weight/day; Cw is the concentration of PAEs in drinking water, µg/L; IR is the ingestion rate, L/day; EF is the exposure frequency, or the number of days of water use per year (365 days); ED is the exposure duration (average life expectancy), years; BW is the average body weight of an adult, kg; and AT is the averaging time, i.e., ED × 365.

From the calculated exposure level (AE) of PAEs, a hazard quotient (HQ) was determined to estimate the non-carcinogenic risk [22], according to the equation:HQ = AE/RfD,(2)
where RfD is the individual reference dose of a phthalate specified by the U.S. EPA [23]. The RfD values for DEP, DBP, BBP, DEHP, and DOP were 800, 100, 200, 20, and 10 µg/kg/day, respectively. In addition, to estimate the total non-carcinogenic risk posed by all PAEs, their respective HQ hazard quotients were summed and expressed as a hazard index (HI). If HI > 1 and/or HQ > 1, this indicates potential adverse health effects [24].

#### 2.7.2. Freshwater Risk Assessment

Risk quotients (RQs) for individual PAEs were used to assess the environmental risk for the three sensitive aquatic species (according to the European Technical Guidelines for Risk Assessment) [25]. The assessment of water risk, expressed as a risk quotient (RQ), was calculated by Formula (3) [26,27,28]:RQ = MEC/PNEC,(3)
where MEC is the measured environmental concentration of PAEs in water (µg/L) and PNEC is the predicted no-effect concentration, i.e., the concentration that is not expected to affect aquatic organisms. PNEC was calculated based on the no observed effect concentration (NOEC) or the median effective concentration (EC_50_), which were divided by the assessment factor (AF) [29]. The average PAE concentrations of three species of hydrobionts (fish, cladocerans, and algae) were used for risk assessment. Toxicity data and assessment factors (AFs) for the six PAEs were taken from the literature and the U.S. Environmental Protection Agency database [30] (Table 1). The calculated risk to the environment can be considered insignificant if RQ < 0.1; low if the RQ is between 0.1 and 1.0; medium if the RQ is between 1 and 10; and high if RQ > 10 [31].

### 2.8. Data Analysis

The results are presented as arithmetic means ± standard deviations (SD). The calculation of means and SDs was carried out in Microsoft Excel. Differences were found to be significant at *p* < 0.05. A Spearman correlation analysis was performed in STATISTICA 13.

## 3. Results

### 3.1. Water Quality

The water quality parameters for the studied lakes are presented in Appendix A. Water pH varied from 6.01 to 7.84, with the highest values in Lakes Bormashevoe and Shchuchye. During the subglacial period, the water temperature in the lakes was within 0.1–0.3 °C, and the turbidity was minimal. The water turbidity in the lakes was as follows (average NTU, in decreasing order): Bormashevoe (22.53) > Arangatui (5.75) > Dukhovoe (2.99) > Kotokel (1.83) > Shchuchye (0.99). Lakes Kotokel, Arangatui, and Dukhovoe had the lowest mineralization, in the range of 58–126 mg/L. Mineralization in Shchuchye was 292 mg/L, and Lake Bormashevoe had the highest value of 1284 mg/L, which is consistent with previous studies [38,39,40,41,42]. Notably, the high mineralization of Lake Bormashevoe could be the result of evaporative concentration, which led to the formation of mineral waters, as noted in [43].

The content of dissolved oxygen in the water of Lakes Kotokel and Shchuchye was 10.8–13.8 mg/L. Close values were previously reported in [38,40] In Lakes Arangatui, Bormashevoe, and Dukhovoe, the content of dissolved oxygen was reduced to 0–2.5 mg/L at certain sampling sites. The contents of easily oxidizable organic matter (PI values) and organic compounds resistant to oxidation (COD values) were highest in Lake Bormashevoe (45.87 and 162.11 mg/L, respectively) and lowest in Lake Shchuchye (4.08 and 13.84 mg/L). Larger COD values (24.0 mg/L) for Lake Shchuchye were reported earlier in [40], but they were sampled during the warm seasons in 2005.

The highest content of nutrients was found in Lake Bormashevoe, especially for mineral and total phosphorus and ammonia nitrogen (1.01, 1.49, and 13.79 mg/L, respectively). These values were an order of magnitude higher (for ammonium, two orders of magnitude higher) than typical contents in other lakes. In Shchuchye, the contents of all measured nutrients were significantly lower.

According to hydrochemical data, the discharge of warm deep fracture-vein water has the strongest effect on the chemical composition of water in Bormashevoe, Arangatui, and Kotokel and a comparatively smaller effect in Dukhovoe. The discharge of these waters brings biogenic substances and elements (phosphorus, nitrogen, potassium, iron, zinc, copper, etc.) to water bodies. In combination with the increased temperature of thermal waters, this leads to the intensive growth of phytoplankton and aquatic vegetation (sapropelic deposits have formed at the bottoms of the water bodies) [43]. Since Bormashevoe is undrained (unlike the other four lakes), it accumulates chemical and biogenic elements and organic matter. It is the only saline lake on the eastern shore of Lake Baikal, the world’s largest freshwater body.

### 3.2. Compositional Profiles and Concentrations of PAEs

All six priority PAEs were found in the water of the studied lakes, with the exception of BBP, which was only found in Shchuchye. The contents of six priority PAEs were measured in the surface water of the lakes during the winter low-water period of 2022. In Lakes Kotokel, Dukhovoe, Bormashevoe, and Arangatui, the total contents of PAEs were 17.34 µg/L, 12.66 µg/L, 20.55 µg/L, and 17.17 µg/L, respectively; the lowest content (2.30 µg/L) was measured in Shchuchye (Table 2).

The main PAEs in lakes Arangatui and Bormashevoe were DEHP (7.41 and 14.45 µg/L, respectively) and DBP (4.29 and 5.45 µg/L, respectively) (Figure 2).

PAEs are listed as priority pollutants by the European Union Commission, China, and Russia [4]. The US EPA considers DEHP a priority pollutant, and the World Health Organization (WHO) has set a recommended value of 8 µg/L for this substance in drinking water [6]. According to U.S. EPA recommendations, DEHP levels below 6 µg/L are considered safe. The approved MACs for PAEs in Russia [44] are shown in Table 2. The current MAC for DBP is set at the level of 200 µg/L (while in 1999 it was equal to 1 µg/L [45], which was 200 times lower than the modern MAC). In Lakes Arangatui and Bormashevoe, phthalates are nonuniformly distributed. For example, the DEHP contents in Bormashevoe varied from 2.83 to 20.81 µg/L (Table 2). Such distributions correlate with the nonuniform chemical compositions of water at the different shores of this lake [46].

The average contents of the other phthalates in all studied lakes did not exceed the MACs established in Russia. DBP was the main phthalate in Lakes Kotokel and Dukhovoe (11.23 and 1.05 µg/L, respectively) (Figure 2).

DMP and DEP are usually used in the production of cosmetics, household goods, and personal care items and are not used in the production of PVC or plasticizers [1,5]. The DMP and DEP contents in the surface water of the lakes were low: up to 1.17 µg/L and up to 2.07 µg/L, respectively (Table 2).

The BBP contents in Arangatui, Bormashevoe, Dukhovoe, and Kotokel were below the detection limits, and in Shchuchye it was detected in surface water samples from three sites out of five in the range of 0.07–0.15 µg/L (Table 2).

The DnOP concentrations in the lakes were insignificant and varied from trace amounts to 0.51 µg/L.

### 3.3. Ecological Risk Assessment

#### Drinking Water Risk Assessment

The calculations of the hazard quotients (HQ) and the total hazard index (HI) for PAEs in the studied freshwater lakes (except for a saline lake, Bormashevoe) are presented in Table 3.

## 4. Discussion

PAEs are usually considered to be industrial pollutants, exogenous substances that are hazardous to humans, but recently there has also been evidence of their biogenic origin. PAEs are reported to possess allelopathic, antimicrobial, insecticidal, and other biological activities, which might enhance the competitiveness of plants, algae, and microorganisms to better accommodate biotic and abiotic stress [20,47]. These compounds have been found in red [48] and freshwater algae, cyanobacteria [49], fungi [50], and higher plants of various families. However, regardless of the origin of PAEs, their high concentrations in aquatic ecosystems may pose a potential hazard to organisms.

According to our measurements, DBP and DEHP were the main PAEs in the studied lakes. The highest concentrations of these pollutants were detected in Arangatui and Bormashevoe, which were under minimal anthropogenic load. The sum proportion of DEHP and DBP in the total content of the six PAEs ranged from 90.7% to 96.6% for four lakes and was 79.1% for Shchuchye. Significant levels of DBP and DEHP in the lakes of the eastern coast of Baikal with low and medium anthropogenic loads suggest a biogenic origin. This assumption is also supported by two points: first, the average positive correlation between the DEHP content and the levels of turbidity, nitrite nitrogen, phosphate ion, and total phosphorus (Appendix A) and, second, the significantly lower concentrations of DEHP and DBP (1.48 µg/L and 0.34 µg/L, respectively) in a lake with a relatively high anthropogenic load (Shchuchye). A positive correlation of the PAE content with nutrients and suspended organic matter was also noted for Xingkai Lake, China [51]. The highest levels of DEHP were found in Bormashevoe, where thermal waters are discharged, leading to the accumulation of bioactive elements such as phosphorus, potassium, iron, zinc, copper, etc., and the accelerated growth of phytoplankton [46]. In the summertime, the discharge accelerates the warming of the lake, leading to the filling of the entire water column with microalgae, which die off in winter and settle to the bottom of the reservoir. The lake bottom sediments are processed by alkalophilic microorganisms year-round due to the supply of warm water. Similarly, Lake Arangatui is also fed by thermal waters, providing similar conditions for its ecosystem. A number of studies have noted that the measurement of PAE concentrations in natural water bodies should be coupled with an analysis of aquatic vegetation, the main supplier of organic matter in aquatic ecosystems. The biogenic origin of DEHP and DBP in *Undaria pinnatifida*, *Laminaria japonica*, and green alga *Ulva* sp. was confirmed by natural ^14^C measurements. The natural abundance of ^14^C content of DEHP obtained from the same algae was about 50–80% of the standard sample, and the ^14^C content of the petrochemical (industrial) products of DBP and DEHP were below the detection limit [52]. Babenko et al. noted that phytoplankton can serve as a source of biogenic PAEs. The maximum concentrations of DEHP were registered in Baikal water samples during periods of ice cover and the development of planktonic algae. In plankton samples collected simultaneously with water samples, the DEHP content reached 1–56 mg/kg of dry weight. In a model experiment in the biomass of S. acus subsp. radians (the dominant species of Baikal phytoplankton) the DEHP content was 5.60 mg/kg of dry weight [53].

The entry of PAEs into water bodies with a low anthropogenic loads can be partially carried out by atmospheric transport, since the stability of PAEs in abiotic conditions is quite high. For example, the half-life of DEHP ranges from 390 to 1600 days due to photolysis and hydrolysis [54]. PAEs have even been detected in remote areas such as the Atlantic, the Arctic Ocean [55], and the Amazon rainforest due to atmospheric transport [56]. For example, the average PAE concentration in PM_2_._5_ in samples collected at Mount Tai, a high-elevation mountain site in northern China, was 19.48 ng/m^3^. The major PAEs in PM_2_._5_ were DEHP, DBP, and DiBP, while DMP and DEP predominated in the gas phase. The PAE concentrations decreased in cloudy or rainy weather and increased again in clear weather [57]. The orographic isolation of the Baikal depression leads to the circulation of air masses over the water area of the lake. This contributes to the long-term preservation of aerosol and gas impurities in the air above the lake, which come both from nearby sources of emissions and as a result of long-range atmospheric transport [58]. The local air circulation over the lake depression causes an exchange of impurities between the water surface and the atmosphere. According to a 1991 study by the Zuev Institute of Atmospheric Optics, high levels of air pollution caused by emissions from the Irkutsk industrial zone and the Baikal Pulp and Paper Mill were detected over the water area [58,59,60,61].

In our opinion, the main sources of PAEs coming due to atmospheric transport are from the Irkutsk-Cheremkhovo industrial hub. These enterprises use technological processes that lead to emissions of organic and inorganic acids, alkalis, toxic chemicals, solvents, dyes, petroleum products, etc. In this area, the winter months are usually unfavorable for the atmospheric dispersion of contaminants: windless conditions as well as surface and elevated atmospheric inversions, while thermal power facilities operate at full capacity. The prevailing northwesterly winds carry large amounts of pollutants from the sources of the Irkutsk-Cheremkhovo depression to the areas of South Baikal along the Angara River. A study using a mathematical model of the spread of pollutants from stationary sources in settlements located in the Baikal Basin showed that specially protected natural areas, including Zabaikalsky National Park, are located in areas of anthropogenic influence [59]. A large amount of plastic is brought to the lakeshores by tourists.

Thus, we can assume that PAEs enter the surface waters of the lakes of the eastern coast of Lake Baikal through two pathways: biogenic and anthropogenic. The biogenic pathway is associated with the plant biosynthesis of PAEs. Meanwhile, the anthropogenic pathway may be related to both the direct input of PAEs from household products used by the local population and atmospheric transport from chemical production facilities.

BBP was only detected in Shchuchye, which suffers from a very high recreational load. Consequently, the presence of BBP may be due to a longer and intensive direct anthropogenic influence. This assumption was also supported by relatively high levels of other phthalates (DMP, DEP, and DnOP) in lakes with high anthropogenic loads: Kotokel (0.37, 0.46, and 0.21 µg/L, respectively) and in Shchuchye (0.02, 0.26, and 0.13 µg/L, respectively). DnOP accounts for more than half (55%) of the phthalates produced in Russia [62]; apparently, it enters the lakes from polymer products used by tourists and the population permanently residing on the lakeshores. For instance, Kotokel is one of the favorite places for tourists from Buryatia and other regions. There are four settlements and about 40 tourist bases and guest houses. The high anthropogenic load probably contributed to the outbreak of Gaff disease in 2008 among the residents of the Lake Kotokel shoreline and tourists. Its symptoms appeared after swimming in Lake Kotokel and eating fish caught there [38]. Restrictions were imposed on visiting and fishing for almost ten years. At present, the restrictive measures have been lifted. However, the above does not exclude the contribution of the atmospheric transport of PAEs from other areas to the lakes.

A number of studies have reported a very wide range of measured PAE concentrations in the waters of the world’s rivers and lakes, from trace values to dozens of µg/L (Table 4). At the same time, regions with high population density and well-developed industry tend to have high contents of PAEs in water.

Based on our data, the contents of PAEs in the lakes of the eastern coast of Lake Baikal varied over a wide range, even within one water body, which indicates their uneven distribution in the water mass. This is due to the geomorphological structures of the lakes, the abundance of microorganisms, zoo- and phytoplankton, the development of aquatic vegetation, and the presence of local sources of pollution. Although PAEs undergo photolysis, biodegradation, and oxidation processes in the environment [69], they can accumulate to levels that are dangerous for living organisms, especially in the subglacial period.

Hazard quotients (HQ) and hazard indices (HI) were determined for the five lakes for each of the PAEs, whose values were significantly lower than unity (Table 3). This indicated no risk of human exposure to PAEs when consuming water from the lakes. Generally, when HQ or HI > 1, it indicates potentially adverse health effects and requires additional monitoring and evaluation [24]. Nevertheless, the PAE concentrations we measured were high enough to pose a threat to sensitive aquatic organisms. To assess aquatic toxicity, we calculated risk quotients (RQ) using the average concentrations of each individual PAE in the surface water of the lakes. Because the calculated RQ < 1, the measured concentrations of DMP, DEP, and BBP posed no or very low risk to the aquatic environment for fish, crustaceans, and algae, which was consistent with the available data [26,28,70] (Figure 3). The potential adverse impact on aquatic organisms, according to our calculations, may be more significant.

Almost all water samples taken in the lakes were classified as medium- or high-risk in terms of DEHP’s impact on aquatic organisms. The calculated risk of DBP exposure to aquatic organisms in Shchuchye was low, and it was medium or high in the other lakes. PAEs were ranked in order of decreasing environmental risk to the aquatic environment: DEHP > DBP > DnOP > BBP > DMP > DEP. A similar ranking was shown for the Pearl River estuary in China [28].

## 5. Conclusions

The contents of six priority PAEs in the surface water of the largest lakes on the eastern shore of Lake Baikal during the winter low-water period were studied for the first time. The low water temperature and low dissolved oxygen in the water in winter contributed to the accumulation of PAEs.

The presence of PAEs in lakes with different ecological statuses suggests that the origin of these pollutants is not only anthropogenic but also biogenic. The biogenic origin of PAEs is associated with aquatic vegetation synthesizing DEHP and DBP in adaptation reactions to abiotic and biotic environmental factors. The anthropogenic origin is associated with both direct influx from the lakes’ catchment area from PAE-containing products used by local people and tourists as well as atmospheric transport from nearby industrial areas. The obtained results confirmed the complex formation of a phthalate pool in the water of lakes located on the eastern shore of Lake Baikal and may represent important information about the distribution of PAEs in the Baikal natural area.

The hazard quotients (HQs) calculated for each phthalate, as well as the total hazard indices (HIs), were well below unity, indicating that there is no risk of human exposure to PAEs when consuming the water from the lakes. However, the calculated risk quotients (RQs) for aquatic organisms showed that high levels of DEHP and DBP in the lakes’ water (as opposed to the other measured phthalates) could potentially have serious adverse effects on the aquatic ecosystem.

The first data on the contents of PAEs in the surface waters of the lakes located on the eastern shore of Lake Baikal indicate the need for further research on the phthalate content and its seasonal dynamics in the entire ecosystem, including the bottom sediments and biota.

## Figures and Tables

**Figure 1 ijerph-20-01173-f001:**
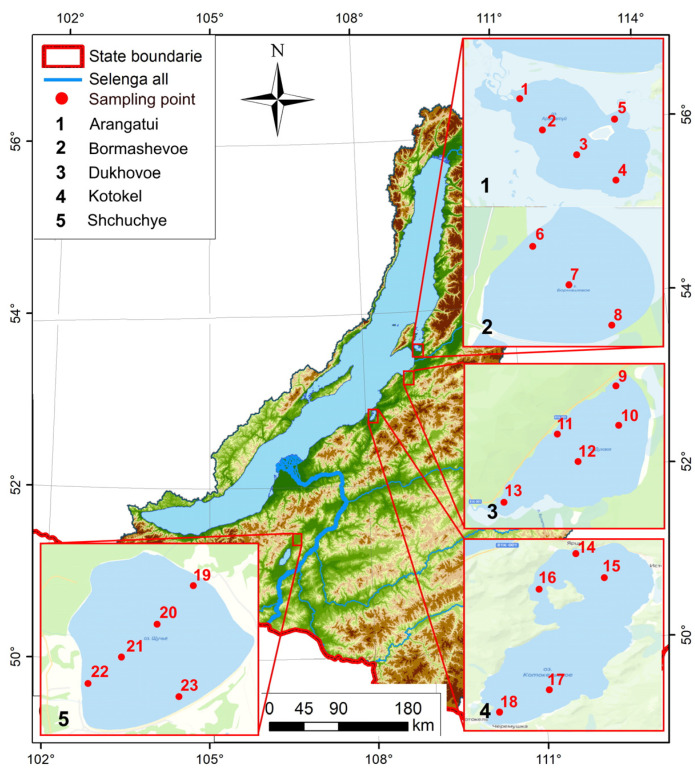
Map of surface water sampling sites in the lakes on the eastern shore of Lake Baikal.

**Figure 2 ijerph-20-01173-f002:**
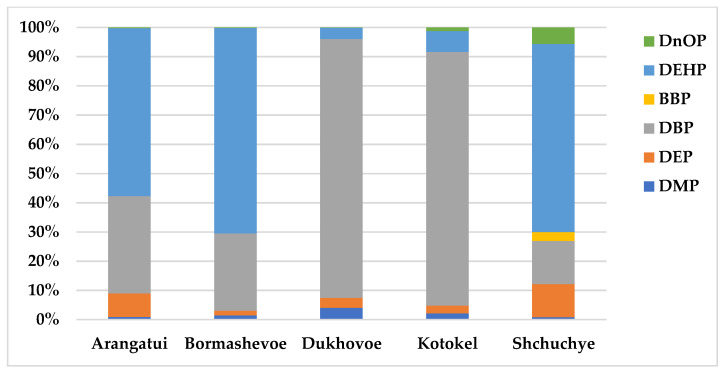
PAE ratios in the surface water of the lakes on the eastern shore of Lake Baikal.

**Figure 3 ijerph-20-01173-f003:**
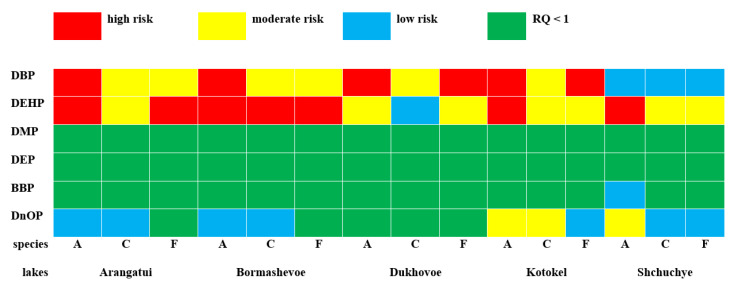
Risk quotients (RQs) determined for the six PAEs found in the lake water for the most sensitive aquatic organisms (A, C, and F—algae, cladocerans, and fishes, respectively).

**Table 1 ijerph-20-01173-t001:** PAE toxicity for the most sensitive aquatic organisms.

PAE	Hydrobiont	Species	Toxicity, µg/L	Assessment Factor, AF	PNEC (Water), µg/L	Reference
DMP	Algae	*Pseudokirchneriella subcapitata*	EC_50_ = 142,000	1000	142	[32]
	Cladoceran	*Daphnia magna*	EC_50_ = 33,000	1000	33	
	Fish	*Lepomis macrochirus*	EC_50_ = 50,000	1000	50	
DEP	Algae	*Pseudokirchneriella subcapitata*	EC_50_ = 16,000	1000	16	
	Cladoceran	*Daphnia magna*	EC_50_ = 86,000	1000	86	
	Fish	*Lepomis macrochirus*	NOEC = 1650	100	16,5	
DBP	Algae	*Pseudokirchneriella subcapitata*	EC_50_ = 400	1000	142	
	Cladoceran	*Daphnia magna*	EC_50_ = 33,000	1000	33	
	Fish	*Danio rerio (Zebra danio)*	NOEC = 100	100	1	[33]
BBP	Algae	*Pseudokirchneriella subcapitata*	NOEC = 30	100	0.3	[34]
	Cladoceran	*Daphnia magna*	EC_50_ = 3700	1000	3.7	[35]
	Fish	*Lepomis macrochirus*	EC_50_ = 1700	1000	17	[32]
DEHP	Algae	*Pseudokirchneriella subcapitata*	EC_50_ = 100	1000	0.1	
	Cladoceran	*Daphnia magna*	EC_50_ = 77	1000	0.77	[34]
	Fish	*Lepomis macrochirus*	EC_50_ = 200	1000	0.2	[32]
DnOP	Algae	*Pseudokirchneriella subcapitata*	EC_50_ = 100	1000	0.1	
	Cladoceran	*Molluscs Haliotis diversicolor*	NOEC = 17.9	100	0.179	[36]
	Fish	*Channel Catfish*	EC_50_ = 700	1000	0.7	[37]

**Table 2 ijerph-20-01173-t002:** PAE levels in the surface water of the lakes located on the eastern shore of Lake Baikal, µg/L.

PAE		MAC [44]	Lakes				
			Arangatui	Bormashevoe	Dukhovoe	Kotokel	Shchuchye
DMP	min-max ^1^ mean	300	nd ^2^–0.23 0.12	nd–0.52 0.30	0.13–0.94 0.51	0.03–1.17 0.37	nd–0.09 0.02
DEP	min-max mean	3000	0.21–2.07 1.04	0.01–0.48 0.32	0.13–0.61 0.43	0.21–0.70 0.46	0.15–0.42 0.26
DBP	min-max mean	200	nd–8.76 4.29	0.95–14.02 5.45	2.53–14.73 11.23	nd–25.50 15.05	nd–1.62 0.34
BBP	min-max mean	–	nd	nd	nd	nd	nd–0.15 0.07
DEHP	min-max mean	8	2.50–18.48 7.41	2.83–20.81 14.45	nd–1.49 0.49	nd–4.98 1.25	nd–4.35 1.48
DnOP	min-max mean	1600	nd–0.10 0.03	nd–0.08 0.03	nd–0.04 0.01	nd–0.51 0.21	nd–0.46 0.13
∑PAEs			12.9	20.6	12.7	17.3	2.3

^1^ The concentration interval is represented in the numerator, the mean value is represented in the denominator. ^2^ nd: not detected or below the MQL; MDLs were within 0.10–0.7 ng/L.

**Table 3 ijerph-20-01173-t003:** AE and HQ values for phthalates, calculated for human consumption of water from the lakes.

Lake	PAE	Level of Exposure (AE), µg/kg/Day	RfD, µg/kg/Day [23]	HQ
**Arangatui**	DBP	0.2942	100	0.00294
	DEHP	0.2541	20	0.01270
	DEP	0.0357	800	0.00004
	BBP	-	200	-
	DnOP	0.0010	10	0.00010
	HI			0.01578
**Bormashevoe**	DBP	0.1869	100	0.00187
	DEHP	0.4954	20	0.02477
	DEP	0.0110	800	0.00001
	BBP	-	200	-
	DnOP	0.0010	10	0.00010
	HI			0.02675
**Dukhovoe**	DBP	0.3850	100	0.00385
	DEHP	0.0168	20	0.00084
	DEP	0.0147	800	0.00002
	BBP	-	200	-
	DnOP	0.0003	10	0.00003
	HI			0.00474
**Kotokel**	DBP	0.5160	100	0.00516
	DEHP	0.0429	20	0.00214
	DEP	0.0158	800	0.00002
	BBP	-	200	-
	DnOP	0.0072	10	0.00804
	HI			0.00313
**Shchuchye**	DBP	0.0117	100	0.00012
	DEHP	0.0507	20	0.00254
	DEP	0.0089	800	0.00001
	BBP	0.0024	200	0.00001
	DnOP	0.0045	10	0.00045
	HI			0.00313

**Table 4 ijerph-20-01173-t004:** PAE contents in the water of the world’s lakes (µg/L).

Lake	DMP	DEP	DBP	DEHP	BBP	DnOP	Reference
Taihu (China)	nd–1.32	0.08–4.79	nd–2.54	nd–1.41	0.08–4.72	0.07–0.590	[63]
Taihu (China)							[64]
*Dry Season*	nd–0.16	nd-0.15	nd–0.48	nd–1.47	nd	nd	
*Normal Season*	nd–0.80	nd–0.12	nd–0.19	nd–3.31	nd–1.31	nd–0.65	
*Wet season*	nd–0.11	nd–0.14	0.02–2.88	nd–2.65	nd–0.68	nd	
Chao Hu (China)	0.015–3.670	0.006–0.283	0.070–17.529	nd–0.576	nd-0.107	nd–0.045	[37]
Dong Hu (China)	—	nd	9	14	—	—	[65]
Da Ming Hu (China)	—	nd	51	8	—	—	[65]
Kunming (China)	—	nd	17	nd	—	—	[65]
Shichahai (China)	0.047–0.143	0.006–0.013	0.009–0.157	0.140–0.519	nd–0.512	0.015–0.022	[66]
Lakes in Summer Palace (China)	0.039–0.082	nd-0.011	0.058–0.515	0.139–0.393	nd–0.021	0.016–0.024	[66]
Baikal (Russia)	—	—	0.35–0.89	0.06–0.32	—	—	[16]
Small Xingkai (China)	0.001–0.011	0.002–0.007	0.104–0.530	0.123–3.247	0.001–0.005	nd–0.04	[51]
Large Xingkai (China)	0.003–0.026	0.003–0.018	0.109–0.520	0.223–3.456	nd-0.002	nd–0.007	
Victoria (Uganda)	0.006–0.400	0.038–1.100	0.35–16.0	0.21–23.0	—	—	[67]
Asan (Korea)	nd–0.18	nd–0.05	nd–0.34	nd–1.34	nd	nd–0.02	[21]
Thien Quang, Ba Mau, Hoan Kiem, Yen So, Truc Bach, and West Lake (Vietnam)	0.115–2.950	0.639–14.00	0.783–34.00	1.040–48.70	0.182–21.100	nd–7.310	[68]

nd: concentration below the MQL; “—“: was not measured.

## Data Availability

Not applicable.

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
