# Peer review of "First Data on PAE Levels in Surface Water in Lakes of the Eastern Coast of Baikal"

_ijerph, 2023, doi:10.3390/ijerph20021173_

Round 1
Reviewer 1 Report
The paper is an excellent example of undertaking a "base-line" investigation into existing PAE concentrations in Lake Baikal. The approach and analysis methods were sound, and the results provide a comparative platform for future monitoring and results interpretation with respect to anthropogenic and biogenic inputs.
I strongly support this scientific approach to obtaining water quality data, gaining insight into sources, patterns, and processes, thus "characterising" the landscape and waters of Lake Baikal. Future variations in water quality can be used to improve land/lake management. Great paper!
Author Response
We thank the Reviewer for their appreciation of our manuscript.
On New Year’s Eve, we wish you all the best! The Rabbit is a symbol of the upcoming 2023 year on the eastern calendar. Happy New Year!
Reviewer 2 Report
Review for the paper Bazarsadueva S V., at al. “First Data on PAE Levels in Surface Water in Lakes of the Eastern Coast of Baikal”.
Such an interesting research object is chosen in the reviewed paper as РАЕs in the water of lakes on the eastern shore of Lake Baikal. The lakes are situated in tectonic depressions formed by rift processes, and their waters chemical composition considerably depends on the income of thermal water. It is important that the lakes are notable by the volume of anthropogenic impact. While studying five lakes at 23 sampling point, 69 samples were collected. The sampling was performed during one month (in February, 2022) and thus, the results of the analysis present the РАЕs content in the water at sampling moment. In principle, this result is presented in the paper title “First Date on PAE Levels …..” but it gives no evidence for a wide discussion, in particular, about biogenic and abiogenic sources, it only opens a way for further interesting research. If РАЕs monitoring had been done for several seasons (summer, winter), the results obtained would serve as a solid base for discussion and for more evident conclusions. Shortening of Discussion section does not decrease the level of the paper in question presenting a peculiar research result.
РАЕs are very complicated analysis objects in surface waters, in particular at their contents at trace levels. Minimal level of determined РАЕs concentrations does not depend on the sensitivity of a method used but it is determined by their level in laboratory backgrounds. The selected by authors methods such EPA Methods 3535 and 8061A are not oriented to the analysis of РАЕs in surface waters at analytes contents at trace levels. The method is characterized by presence of numerous stages while samples preparation, and at each of them (filtering, SPE, concentrating of eluates at rotor evaporator and in nitrogen flux), the risk of secondary contamination of samples is very great. The amount of РАЕs in laboratory backgrounds can be compared to their content in the studied water sample (see Table S4, Table 2). It is necessary to notice that the reviewed paper is not aimed to development of a method for РАЕs determination in water at trace level, however, the correctness and accuracy of РАЕs determination are very important for determination of the reliability of the presented results and of the article value in the whole. In the manuscript presented, the given characteristics are not sufficient for such an assessment.
The section Ecological risk assessment is very valuable scientifically and very important for the discussion of the research results taking into account relatively high (relatively to background concentrations) РАЕs concentrations found in the lakes water.
It is necessary to pay attention to the following methodological moments:
1. РАЕs concentrations were calculated by distraction from the results of measurement of average values of the field and laboratory background samples (procedural blanks?). How did the authors assess «procedural blanks» (Table S4)?
2. The authors performed graduation in the concentrations range from 1.0 to 1000 µg/L while in the studied water samples, РАЕs content did not exceed 25.5 µg/L. It is evident that it was necessary to perform the graduation in the expected range of РАЕs concentrations in water samples.
3. How did the authors calculated РАЕs concentrations? Did they use method of absolute graduation, using linearity curve of stated by chromatography of prepared standard solutions?
4. Were deuterated surrogate standards DMP-d4, DEHP-d4 used for the assessment of recovery of the methods? Recovery range is assessed as a rather great value, up to 30%. This result was taken accounts in the calculation of final PAE concentrations?
Author Response
REVIWER 2
We thank the Reviewer for their time with our manuscript and for sharing the review. Indeed, phthalates are challenging objects for analysis, so we tried to take into account all analytical factors that could affect the results of the measurements. Our study of phthalates in water bodies in the Baikal Natural Area we began by reviewing relevant EPA regulations. Then we reviewed similar studies and took their experience into account. This helped us to minimize possible methodological difficulties. In this work, for the first time, we determined the content of phthalates in the studied lakes, and outlined directions for further research.
- РАЕs concentrations were calculated by distraction from the results of measurement of average values of the field and laboratory background samples (procedural blanks?). How did the authors assess «procedural blanks» (Table S4)?
We thank the reviewer for the question. As you rightly pointed out, this is not an accurate translation — it should be “procedural blanks” instead of “laboratory background samples”.
According to IUPAC, “…a procedural blank is a sample that does not contain the matrix, that is brought through the entire measurement procedure and analysed in the same manner as a test sample”.
Procedural blanks were treated as described in the Section 2.3., with the difference that deionized water was used instead of lake water.
“Deionized water sample (500 mL) were filtered through a 0.45 µm fiberglass membrane in a glass filter unit. The PAEs were concentrated by solid phase extraction (SPE) on ENVI-18 cartridges (500 mg, 6 mL, Supelco, USA) using a VM12 vacuum manifold (Phenomenex, USA), which were sequentially conditioned with 5 mL acetone, 5 mL methylene chloride, 5 mL ethyl acetate, 5 mL acetone, and 5 mL purified water. The cartridges were then dried for 20 minutes, and the PAEs were eluted sequentially with 6 mL of methylene chloride and 6 mL of n-hexane at a rate of 1 mL/min [Young-Min Lee, Jung-Eun Lee, Wooseok Choe, Taeyeon Kim, Ji-Young Lee, Younglim Kho, Kyungho Choi, Kyung-Duk Zoh, Distribution of phthalate esters in air, water, sediments, and fish in the Asan Lake of Korea //Environment International. – 2019. –Vol. 126. – P. 635-643, https://doi.org/10.1016/j.envint.2019.02.059]. The solvents from the eluate were removed under vacuum at a rotary evaporator to a volume of 1 mL, then evaporated almost to dryness under a weak flow of nitrogen. The residue was dissolved in 1 mL of n-hexane for analysis by gas chromatography–mass spectrometry (GC/MS)”.
- The authors performed graduation in the concentrations range from 1.0 to 1000 µg/L while in the studied water samples, РАЕs content did not exceed 25.5 µg/L. It is evident that it was necessary to perform the graduation in the expected range of РАЕs concentrations in water samples.
We thank the reviewer for the important question. The concentration values initially calculated in Agilent MassHunter Quantitative analysis (Quantitative analysis of the environment (MS)) were further re-calculated as follows:
X=C*k,
where C – the concentration calculated in the program,
k – the coefficient accounting for sample concentration/dilution in the process of sample preparation.
In our study, all samples ready for GC/MS analysis were divided into two parts. The first parts were used to determine concentrations. If the concentration range was exceeded for some samples, we used the second part while reducing the volume of the sample injected into the evaporator.
- How did the authors calculated РАЕs concentrations? Did they use method of absolute graduation, using linearity curve of stated by chromatography of prepared standard solutions?
Thank you for the question. Yes, the phthalate concentrations were calculated using the absolute graduation method in the program Agilent MassHunter Quantitative analysis (Quantitative analysis of the environment (MS)).
- Were deuterated surrogate standards DMP-d4, DEHP-d4 used for the assessment of recovery of the methods? Recovery range is assessed as a rather great value, up to 30%. This result was taken accounts in the calculation of final PAE concentrations?
Thank you for the question. Yes, deuterated surrogate standards DMP-d4, DEHP-d4 were used for the assessment of recovery of the methods. The obtained recovery values were taken into account when calculating the final phthalate concentrations.
On New Year’s Eve, we wish you all the best! The Rabbit is a symbol of the upcoming 2023 year on the eastern calendar. Happy New Year!
We hope that the Reviewer will find our responses to the comments satisfactory, and the revised version of the manuscript will be accepted.
Your sincerely,
Selmeg V. Bazarsadueva
Reviewer 3 Report
The paper studies some data regarding PAE level in surface water. Overall, the paper is written well, however, few comments should be considered before accepting the paper for publication.
1. The abstract needs to be shorten.
2. The novelty of the study should be highlighted better in the last paragraph of the introduction.
3. Lines 106-123 need to be summarized better.
4. section 3.1: water quality results need to be validated with previouse study.
Author Response
Reviewer 3
We are grateful to Reviewer for their work with our article and the comments provided.
- The abstract needs to be shorten.
The Abstract has now been shortened; its length is 197 words. The abstract is shorten as follows:
Abstract: The increasing consumption of phthalates (PAEs) along with their high toxicity and high mobility poses a threat to the environment. This study presented initial data on the content of six priority PAEs in the water of lakes located on the eastern shore of Lake Baikal-Arangatui, Bormashevoe, Dukhovoe, Kotokel, and Shchuchye. Mean total concentrations of 6 PAEs in lakes Arangatui and Bormashevoe (low anthropogenic load) were comparable to those in Kotokel (medium anthropogenic load, 17.34 µg/L), but were significantly higher (P < 0.05) than in Dukhovoe and Shchuchye (high anthropogenic load, 10.49 and 2.30 µg/L, respectively). DBP and DEHP were the main PAEs in all samples. The DEHP content in lakes Arangatui and Bormashevoe was quite high, and at some sampling sites it exceeded the MACs established by Russian, U.S. EPA and WHO regulations. The assessment showed that there is no potential risk to humans associated with the presence of PAEs in drinking water. However, the levels of DEHP, DBP and DnOP in the water pose a potential threat to sensitive aquatic organisms, as shown by the calculated risk quotients (RQ). It is assumed that the origin of phthalates in the studied lakes is both anthropogenic and biogenic.
- The novelty of the study should be highlighted better in the last paragraph of the introduction.
We highlighted the novelty in this paragraph:
“For the first time, this paper reported the initial data on the contents of 6 priority PAEs in the ecosystems of large lakes (Arangatui, Bormashevoe, Dukhovoe, Kotokel, and Shchuchye, located on the eastern shore of Lake Baikal) which have varying degrees of anthropogenic load. The results obtained confirm the complex formation of the phthalate pool in the water of the lakes, and may represent important information about the distribution of PAEs in the Baikal natural area.”
- Lines 106-123 need to be summarized better.
Thank you for this advice, it has been summarized and should sound much better now:
Their average depth range from 2 to 6 m, and the area varies from 1.3 to 62.9 km2. The chemical composition of the water in these lakes is significantly influenced by the discharge of thermal fracture–vein waters. Two lakes—Shchuchye and Kotokel—have long been intensively used as recreational water bodies, where dozens of camping sites and guest houses are located. The territory where the lakes are located is characterized by a sharply continental climate, the average annual temperatures are negative and range from −3 °C to −2 °C, a short period of frost–free days (95–100 days), and relatively low precipitation ranging from 200 to 350 mm (50% of which falls in July and August and only 15% in the cold season). Due to severe cold winters (average monthly air temperature remains negative for 6 months) lakes freeze to a considerable depth (ice thickness up to 1.2 m), and during the warm season there is a short ice-free period. Detailed information about the geographical and hydrological characteristics of the studied lakes and the anthropogenic load data were provided in Appendix A. Supplementary data. The selected lakes are convenient models for comparative analysis of phthalate pool formation under different conditions.
- Section 3.1: water quality results need to be validated with previouse study.
We validated our data with previous studies.
3.1. Water quality
The water quality parameters for the studied lakes are presented in Table S5. Water pH varied from 6.01 to 7.84, with the highest values in Lakes Bormashevoe and Shchuchye. During the subglacial period, the water temperature in the lakes was within 0.1–0.3 °С, and its turbidity was minimal. Water turbidity in the lakes was as follows (average NTU, in decreasing order): Bormashevoe (22.53) > Arangatui (5.75) > Dukhovoe (2.99) > Kotokel (1.83) > Shchuchye (0.99). Lakes Kotokel, Arangatui and Dukhovoe had the lowest mineralization in the range of 58–126 mg/L, Shchuchye — 292 mg/L, and Lake Bormashevoe had the highest value — 1284 mg/L. which is consistent with previous studies [39-43]. Notably, the high mineralization of Lake Bormashevoe could be the result of evaporative concentration, which led to the formation of mineral waters, as noted in [44].
The content of dissolved oxygen in the water of Lakes Kotokel and Shchuchye was 10.8–13.8 mg/L. Close values were previously reported in [39] и [41] In Lakes Arangatui, Bormashevoe and Dukhovoe, the content of dissolved oxygen was reduced to 0–2.5 mg/L at certain sampling sites. The content of easily oxidizable organic matter (PI values) and organic compounds resistant to oxidation (COD values) was the highest in Lake Bormashevoe (45.87 and 162.11 mg/L, respectively) and the lowest in Lake Shchuchye (4.08 and 13.84 mg/L). Larger COD values (24.0 mg/L) for Lake Shchuchye were reported earlier in [41], but they were also sampled during the warm seasons in 2005.
The highest content of nutrients was found in Lake Bormashevoe, especially for mineral, total phosphorus and ammonia nitrogen (1.01, 1.49 and 13.79 mg/L, respectively). These values are an order of magnitude higher (for ammonium—two orders of magnitude higher) than typical contents in other lakes. In Shchuchye, the content of all the measured nutrients was significantly lower.
According to hydrochemical data, the discharge of warm deep fracture-vein water has the strongest effect on the chemical composition of water in Bormashevoe, Arangatui and Kotokel, and a comparatively smaller effect in Dukhovoe. Discharge of these waters brings biogenic substances and elements (phosphorus, nitrogen, potassium, iron, zinc, copper, etc.) in water bodies. In combination with the increased temperature of thermal waters, this leads to the intensive growth of phytoplankton and aquatic vegetation (sapropelic deposits have formed at the bottom of the water bodies) [44]. Since Bormashevoe is undrained (unlike the other four lakes), it accumulates chemical, biogenic elements and organic matter. It is the only saline lake on the eastern shore of Lake Baikal, the world’s largest freshwater body.
- Lake Kotokel'skoe: Natural Conditions, Biota, Ecology; Ubugunov, L.L., Pronin, N.M., Eds.; Buryat Scientific Center SB RAS Publishing: Ulan-Ude, Russia, 2013; p. 320.
- Angakhaeva, N.A.; Plyusnin, A.M.; Ukraintsev, A.U.; Chernyavskii, M.K.; Peryazeva, E.G.; Zhambalova, D.I. Hydrogeochemical features of Lake Kotokel. Earth sciences and subsoil use 2021, 44, 106-115, doi:10.21285/2686-9993-2021-44-2-106-115.
- Khakhinov, V.V.; Namsaraev, B.B.; Ul’zetueva, I.D.; Barkhutova, D.D.; Abidueva, E.Y.; Banzaraktsaeva, T.G. Hydrochemical and microbiological characteristics of the Gusino-Ubukunskaya group of water bodies. Water Resources 2005, 32, 73-78, doi:10.1007/s11268-005-0010-7.
- Maltsev, A.E.; Bogush, A.A.; Leonova, G.A. Peculiarities of the chemical composition of pore water in the Holocene sapropel section of Dukhovoe Lake (Southern Baikal). Chemistry for Sustainable Development 2014, 22, 517-534.
- Tsyrenova, D.D.; Garankina, V.P.; Dagurova, O.P.; Dambaev, V.B. Conditions for the cyanobacteria development in the lakes of Baikal coastal zone Buryat State University Bulletin (Chemistry, Physics) 2016, 4, 11-16, doi:10.18101/2306-2363-2016-4-11-16.
- Peryazeva, E.G.; Plyusnin, A.M.; Garmaeva, S.Z.; Budaev, R.T.; Zhambalova, D.I. Peculiarities of the formation of the chemical composition of water in lakes on the eastern shore of Lake Baikal. Nat. Res. 2016, 5, 49-59, doi:10.21782/GIPR0206-1619-2016-5(49-59).
On New Year’s Eve, we wish you all the best! The Rabbit is a symbol of the upcoming 2023 year on the eastern calendar. Happy New Year!
We hope that the Reviewer will find our responses to the comments satisfactory, and the revised version of the manuscript will be accepted.
Your sincerely,
Selmeg V. Bazarsadueva
Reviewer 4 Report
Please find the attachment

Author Response
REVIWER 4
We appreciate the Reviewer’s advice and suggestions regarding our manuscript. We have taken all of your comments into consideration (the responses are provided below). We strongly believe that the manuscript has been further improved after the revisions made according to the Reviewer’s comments mentioned below. We have taken them fully into account when revising it.
Section Title
- L 2: Revise “PAE Levels” to “PAE Level”.
We thank the reviewer for the suggestion. We revised “PAE Levels” to“PAE Level” in the sentence: “First Data on PAE Level in Surface Water in Lakes of the Eastern Coast of Baikal”.
Section Abstract
- L 16: Revise “This study presents” to “This study presented”.
- L 17: Revise “Baikal—Arangatui” to “Baikal-Arangatui”.
We have revised these phrases as follows: “This study presented initial data on the content of six priority PAEs in the water of lakes located on the eastern shore of Lake Baikal-Arangatui, Bormashevoe, Dukhovoe, Kotokel, and Shchuchye.”
- L 20: Revise “p < 0.05” to “P < 0.05”. Note that P here is capitalized and italicized.
Thanks again for advice!
- L 22: Revise “is quite high” to “was quite high”.
- L 22: Revise “it exceeds” to “it exceeded”. Please pay attention to tenses.
We have revised these phrases as follows:” The DEHP content in lakes Arangatui and Bormashevoe was quite high, and at some sampling sites it exceeded the MACs established by Russian, U.S. EPA and WHO regulations.”
- L 23: Regarding “Russian, U.S. EPA and WHO regulations”, please provide references to support the above information.
We have not added missing references to “Russian, U.S. EPA and WHO regulations” to the Abstract. These are presented on L. 306-315: “PAEs are listed as priority pollutants by the European Union Commission, by China and Russia [4]. The US EPA considers DEHP a priority pollutant, and the World Health Organization (WHO) has set a recommended value of 8 µg/L for this substance in drinking water [6]. According to U.S. EPA recommendations, DEHP levels below 6 µg/L are considered safe. The approved MACs for PAEs in Russia [45] are shown in Table 2.
- Net, S.; Sempéré, R.; Delmont, A.; Paluselli, A.; Ouddane, B. 4 Occurrence, Fate, Behavior and Ecotoxicological State of Phthalates in Different Environmental Matrices. Environ. Sci. Technol. 2015, 49, 4019-4035, doi:10.1021/es505233b.
- World Health Organization. Global assessment of the state-of-the-science of endocrine disruptors. Available online: https://apps.who.int/iris/handle/10665/67357 (accessed on 28 November 2022).
- SanPiN 1.2.3685-21 Hygienic Standards and Requirements to Ensure Safety and (or) Harmful for Human Environmental Factors.”
- L 28: Revise “source of origin” to “The Source of origin”.
Thanks again for pointing out this typing error. “Source of origin” is added in Keywords.
Section Introduction
- L 37: Revise “20–60%” to “20-60%”.
We thank the reviewer for the suggestion. “20–60%” is revised to “20-60%”.
- L 53: Revise “Some authors note” to “Some authors noted”.
We have revised these phrase as follows: ”Some authors noted a high degree of plastic pollution in the lake [15].”
- L 54: Revise “the DBP and DEHP content” to “DBP and DEHP contents”.
- L 55: Revise “was determined” to “were determined”.
We have revised these phrases as follows: ” In a recent study, the DBP and DEHP contents in Baikal water were determined to be 0.35–0.89 and 0.06–0.32 µg/L, respectively.”
- L 75: Revise “This paper is aimed” to “This paper was aimed”.
- L 75: Revise “the content of 6 priority ” to “the contents of 6 priority ”.
We thank the reviewer for the suggestions. We have revised these phrases as follows: ” This paper was aimed at obtaining first data on the contents of 6 priority PAEs in the ecosystems of large lakes located on the eastern shore of Lake Baikal, which have varying degrees of anthropogenic load.”
Section Materials and methods
- L 83-84: Please provide country information on “Sigma-Aldrich”.
We thank the reviewer for the suggestion. We indicated country information on Sigma-Aldrich: “The standard PAEs compounds, such as dimethyl phthalate (DMP), diethyl phthalate (DEP), dibutyl phthalate (DBP), benzyl butyl phthalate (BBP), di-n-octyl phthalate (DnOP), and di-(2-ethylhexyl) phthalate (DEHP) as well as deuterated surrogate standards DMP-d4, DEHP-d4 and EPA Phthalate Esters Mix were procured from Sigma-Aldrich (USA).”
- L 88: Revise “μg/ml” to “μg/mL”.
Thanks again for pointing out this typing error. “μg/ml” is revised to “μg/mL: “EPA Phthalate Esters Mix (Accustandard Inc., New Haven, CT, USA) included 2000 μg/mL of each component (DMP, DEP, DBP, BBP, DEHP, and DnOP).”
- L 93: Revise “using anhydrous” to “with anhydrous”.
We have revised “using anhydrous” to “with anhydrous” in sentence: «The eluates were dehydrated with anhydrous sodium sulfate (Lenreaktiv, Russia).”
- L 96: Revise “H2O:H2SO4” to “H2O:H2SO4”. Please pay attention to the subscript
Thanks again for pointing out this typing error. “H2O:H2SO4” is revised to “H2O:H2SO4”.
- L 121: Revise “are provided” to “were provided”.
We have revised the phrase as follows: “Detailed information about the geographical and hydrological characteristics of the studied lakes and the anthropogenic load data were provided in Appendix A. Supplementary data.”
- L 131: Revise “is show” to “were show”.
We thank the reviewer for the suggestion. We have revised the phrase as follows: “A map with the studied lakes and sampling sites were shown in Figure 1.”
- L 176: Revise “is given in the tables (Table S4).” to “were given in Table S4.” Are my changes right?
We agree with all changes and very thank the reviewer for all suggestions. We have revised the phrase as follows: “Additional information about the method, such as reproducibility, accuracy, etc. were given in the table S4.”
- L 185: Revise “by the photometric method using salicylic acid” to “with salicylic acid using the photometric method” Are my changes right?
We have revised the phrase as follows: “Nitrate concentration was determined with salicylic acid using the photometric method, the concentration of ammonium ion by photometric detection using Nessler’s reagent, and phosphate concentration by photometric detection through ascorbic acid deoxidation.”
- L 192: Revise “are presented in” to “were presented in”.
We have revised “are presented in” to “were presented in” in the sentence: “The water quality parameters of the lakes on the eastern shore of Lake Baikal were presented in Table S5.”
- L 241: Revise “were found to be significant at p ≤ 0.05” to “were found to be significant at p < 0.05”. Please reconfirm the level of significant difference you obtained.
We thank the reviewer for pointing it out. It is significant. We have revised the phrase as follows: “Differences were found to be significant at p < 0.05.”
Section Discussion
- L 324: Revise “xenobiotics” to “exogenous substances”.
We thank the reviewer for all suggestions. It is a fair point. We have revised “xenobiotics” to “exogenous substances” in sentence: “PAEs are usually considered to be industrial pollutants–exogenous substances that are hazardous to humans, but recently there has also been evidence of their biogenic origin.”
- L 335: Revise “six PAEs ranges” to “six PAEs ranged”.
We have revised the phrase as follows: “The sum proportion of DEHP and DBP in the total content of the six PAEs ranged from 90.7% to 96.6% for 4 lakes, and 79.1% for Shchuchye.”
- L 395-399: Regrading “Thus, we can assume that PAEs enter the surface waters of the lakes of the eastern coast of Lake Baikal through two pathways—anthropogenic and biogenic. The anthropogenic pathway can be related both to direct input from household products used by the local population living on the shores of Kotokel and Shchuchye, and to atmospheric transport.”, 1), please supplement informatioln on biogenic parthway; 2), “atmospheric transport” is not anthropogenic pathway. Please rewrite above contents.
We have revised the phrase as follows: “Thus, we can assume that PAEs enter the surface waters of the lakes of the eastern coast of Lake Baikal through two pathways—biogenic and anthropogenic. The biogenic pathway is associated with plant biosynthesis of PAEs. Meanwhile, the anthropogenic pathway may be related both to the direct input of PAEs from household products used by the local population, and to atmospheric transport from chemical production facilities”.
- L 402: Revise “This assumption is also supported” to “This assumption was also supported”.
We have revised the phrase as follows: “This assumption was also supported by relatively high levels of other phthalates (DMP, DEP and DnOP) in lakes with high anthropogenic load: Kotokel (0.37; 0.46; 0.21 µg/L, respectively) and in Shchuchye (0.02, 0.26, 0.13 µg/L, respectively).”
- L 415: Revise “studies report” to “studies reported”.
We have revised the phrase as follows: “A number of studies reported a very wide range of measured PAEs concentrations in the waters of the world’s rivers and lakes—from trace values to dozens of µg/L (Table 4).”
- L 425: Revise “varies over” to “varied over”.
We have revised the phrase as follows: “Based on our data, the content of PAEs in the lakes of the eastern coast of Lake Baikal varied over a wide range even within one water body, which indicates their uneven distribution in the water mass.”
- L 434: Revise “indicates no” to “indicated no”.
We thank the reviewer for the suggestion. We have revised the phrase as follows: “This indicated no risk of human exposure to PAEs when consuming water from the lakes.”
Section Conclusions
- L 459: Regarding “Low water temperature and solar insolation levels”, “Low solar insolation level” did not support conclusion, “solar insolation levels” only appeared here one time in the full text.
After the manuscript was translated into English, the important meaning of our assumption was lost. After editing, it looks as follows: “Low water temperature and low dissolved oxygen in the water in winter contribute to the accumulation of PAEs.»
- L 476: Revise “the content of PAEs” to “the contents of PAEs”.
We have revised the phrase as follows: “The first data on the contents of PAEs in the surface waters of the lakes located on the eastern shore of Lake Baikal indicate the need for further research on the phthalate content and its seasonal dynamics in the entire ecosystem, including bottom sediments and biota.”
On New Year’s Eve, we wish you all the best! The Rabbit is a symbol of the upcoming 2023 year on the eastern calendar. Happy New Year!
We hope that the Reviewer will find our responses to the comments satisfactory, and the revised version of the manuscript will be accepted.
Your sincerely,
Selmeg V. Bazarsadueva